# Association of Updating Identification Documents with Suicidal Ideation and Attempts among Transgender and Nonbinary Youth

**DOI:** 10.3390/ijerph19095016

**Published:** 2022-04-20

**Authors:** Jonah P. DeChants, Myeshia N. Price, Amy E. Green, Carrie K. Davis, Casey J. Pick

**Affiliations:** The Trevor Project, P.O. Box 69232, West Hollywood, CA 90069, USA; myeshia.price@thetrevorproject.org (M.N.P.); amy.green@thetrevorproject.org (A.E.G.); carrie.davis@thetrevorproject.org (C.K.D.); casey.pick@thetrevorproject.org (C.J.P.)

**Keywords:** LGBTQ youth mental health, transgender youth, nonbinary youth, identification documents, suicide

## Abstract

This study examines the association of access to concordant identity documents with attempting suicide in the last year among transgender and nonbinary youth. Data came from 6581 transgender and nonbinary youth who completed an online survey of lesbian, gay, bisexual, transgender, or queer (LGBTQ) youth ages 13–24 residing in the United States. Multivariate logistic regression was used to determine the adjusted odds of attempting suicide in the past year based on whether or not youth were able to change their identification documents. Both wanting to update one’s documents but not being able to (aOR = 2.04, CI = 1.412–2.95; *p* < 0.001) and being able to update one’s documents but not having done so (aOR = 1.74, CI = 1.22–2.50; *p* < 0.001) were associated with greater odds of attempting suicide in the last year, compared to youth who had changed their documents. Revising the way gender is captured on legal documents may be an efficacious public health intervention to reduce suicide risk among transgender and nonbinary youth.

## 1. Introduction

For many transgender and nonbinary people, using identification documents that do not match their gender identity and/or expression can disclose their identity to other people and expose them to harassment and violence [1]. When using documents not concordant with their gender identity or expression, 40% of transgender adults report harassment and 15% report being asked to leave a business or public accommodation [2]. The stress of having one’s gender identity non-consensually disclosed and the possibility of harassment and violence contribute to minority stress among transgender and nonbinary individuals [3]. This minority stress is, in turn, associated with mental health disparities, including elevated risk of anxiety, depression, and suicidality [4,5].

The process of updating one’s identification documents to have a new name or gender marker is time-consuming, logistically challenging, and expensive—when it is legally possible at all. Not all transgender and nonbinary people who desire to change their documents have the resources to do so. Globally, few countries offer the option to update one’s gender marker on identification documents, especially for youth. However, Ireland’s Ombudsman for Children has argued that children’s right to their gender identity is covered by the United Nations’ (UN) Convention on the Rights of the Child under “children’s right to identity” [6]. Argentina has also granted youth under age 18 the right to update their documents under the Gender Recognition Law of 2012 [6]. 

In the United States (U.S.), updating identification documents is complicated by the variation in legal procedures required to update the documents across different states within the U.S. Requirements for updating identification documents vary by jurisdiction and can include submitting a letter from a medical provider or publishing a notification of a name change in a newspaper [7,8]. The requirement of having surgery in order to update one’s identification documents is another barrier for transgender and nonbinary people who may have no interest in, or access to, surgery. The problem is compounded for nonbinary or intersex people, who frequently only have the option to choose “M” or “F” to designate their gender. Only 22 states currently allow nonbinary residents to designate their gender with an “X” on their driver’s license [7]. 

There is a growing body of scientific literature examining the effects of changing one’s identification documents to affirm gender identity among transgender adults. While legal name change is just one part of achieving gender concordance in identity documents, studies of the effects of legal name change have found positive associations with well-being. Among transgender women of color, completing a legal name change has been found to be associated with higher monthly income and higher likelihood of renting or owning a home in the last year [9]. Individuals who had not completed their legal name change were more likely to delay medical care due to their gender identity, use non-prescribed hormones injected by friends, or experience verbal harassment from family and friends [9]. A recent U.S.-based cross-sectional study by Scheim and colleagues found that transgender adults who have updated identification documents report lower rates of serious psychological distress, suicidal ideation, and suicidal planning, compared to those with no gender-concordant documents [10]. Having some, but not all, gender-concordant identification documents is also associated with lower rates of suicidal ideation [9]. 

Transgender adults who have changed their legal gender on both their passport and driver’s license report lower odds of experiencing emotionally upsetting responses due to gender-based mistreatment [11]. Those who had updated their gender marker on just one document had approximately half the odds of experiencing an emotionally upsetting response due to gender-based mistreatment, compared to those who had not updated any [11]. These studies suggest that access to gender-concordant identification documents is associated with positive indicators of transgender adults’ health and economic stability. 

The current study aims to fill a gap in the literature pertaining to gender-concordant identification documents among transgender and nonbinary youth, using many of the same variables as Scheim and colleagues’ 2020 study of the mental health associations of gender-concordant identification documents among transgender adults [10]. Scheim and colleagues’ study was chosen due to the similarity of both outcome and predictor variables in our dataset and the desire to see if similar findings could be replicated among transgender and nonbinary youth. Specifically, we examined the adjusted odds of attempting suicide in the past year among transgender and nonbinary youth who had updated their identification documents, who wanted to update their documents but were not able to, and who were able to update their documents but had not yet done so, while adjusting for associations of age, gender, sexual identity, race/ethnicity, income, census region, immigration status, pronoun use, use of gender-affirming hormones, and parental gender-identity support. 

## 2. Materials and Methods

### 2.1. Procedure

A nonprobability sample of 34,759 self-identified lesbian, gay, bisexual, transgender, or queer (LGBTQ) youth ages 13–24 who resided in the United States was collected online from October 2020 to December 2020. Youth participants resided across each of the 50 U.S. states. To conform with U.S. regulations for online data collection, 13 was set as the minimum age and, to align with the World Health Organization’s definition of “young people”, 24 was set as the maximum age [12,13]. Potential participants were recruited via targeted ads on Facebook, Instagram, and SnapChat. Participants were defined as being LGBTQ if they identified with a gender identity other than cisgender, a sexual identity other than heterosexual, or both. In order to ensure racial and gender diversity in the survey sample, sample quotas that reflected expected rates of youth who were assigned either female or male at birth across race/ethnicity were created and, once those quotes were met, youth from overrepresented groups were pathed out of the survey. Qualified participants completed an online questionnaire that included a maximum of 142 questions, including questions on gender identity. Only the subset of youth who identified as transgender or nonbinary are included in the current study. All research materials were reviewed and approved by an independent Institutional Review Board, Solutions IRB. Solutions IRB is fully accredited by the Association for the Accreditation of Human Research Protection Programs. Youths’ participation in the survey was voluntary, and informed consent was obtained. A waiver of parental consent for youth aged 13–17 years was obtained from the IRB, as the research posed a minimal risk to participants and could have presented potential harm for youth who were not out to their parents about their LGBTQ identity. No names or personal details were included in the survey to ensure anonymity. 

### 2.2. Measures

#### 2.2.1. Identification Documents 

Youths’ ability and interest in updating their identification documents was assessed via a survey item which asked, “Are you able to change your official documents (like your driver’s license or birth certificate) to match your gender identity where you live?”. Response options included (1) No, and I don’t want to change them, (2) No, but I would like to change them if I was able, (3) Yes, and I want to change them when I am able, (4) Yes, I have changed them, (5) Yes, but I don’t want to change them, and (6) I don’t know. Youth who reported that they did not want to change their documents or did not know if they wanted to change their documents were excluded from analysis, since they do not currently need access to updated identification documents. A three-group variable was created to compare (1) youth who had updated their documents, (2) youth who wanted to update their documents but were not able to, and (3) youth who were able to update their documents but had not yet done so.

#### 2.2.2. Suicide Risk 

The item assessing attempting suicide in the last year was taken from the Centers for Disease Control and Prevention (CDC)’s Youth Risk Behavior Survey (YRBS) [14]. Youth were asked “During the past 12 months, how many times did you actually attempt suicide?” Responses were dichotomized with none coded as “0” and one or more coded as “1”. This item has been found to have good convergent and discriminant validity in a previous study [15]. 

#### 2.2.3. Sociodemographic Covariates

Sociodemographic covariates were selected to align with the Scheim and colleagues’ 2020 study of the relationship between having gender-concordant documents and mental health outcomes among transgender adults [10]. Gender identity was measured using a two-stage question of gender identity and sex assigned at birth [16]. Respondents were first asked “What sex were you assigned at birth, on your original birth certificate?” with response options including (1) Male, and (2) Female. Respondents were then asked “Which of the following terms best describes your current gender identity? We understand that there are many different ways youth identify, please pick the one that best describes you here first.” with response options including (1) Girl or woman, (2) Boy or man, (3) Nonbinary, genderfluid, or gender non-conforming, (4) I am not sure or questioning, or (5) I don’t know what this question means. 

Sociodemographic variables included age, gender identity (transgender girl/woman, transgender boy/man, nonbinary assigned male at birth, nonbinary assigned female at birth), sexual identity (gay/lesbian, bisexual, pansexual, queer, questioning, or heterosexual), race/ethnicity (Alaska Native/American Indian, Asian/Pacific Islander, Black/African American, Latinx, multiracial, or White), socioeconomic status (just able to meet basic needs or less, more than able to meet basic needs), U.S. Census Region (Northeast, South, Midwest, West), and whether or not the youth was born in the U.S. 

#### 2.2.4. Gender Identity Related Covariates 

Access to gender-affirming hormone treatment (GAHT) was assessed by a question asking youth “Are you currently taking gender-affirming hormones?” with response options that included (1) “No, and I don’t want to take them,” (2) “No, but I would like to take them,” and (3) “Yes.” Parental support of youths’ gender identity was assessed using a question asking youth “Do you have at least one parent who is supportive of your gender identity?” with answers of (1) “No,” (2) “Yes,” and (3) “I am not ‘out’ about my gender to any of my parents.” Pronoun use was assessed by asking “What pronouns do you use?”. Multiple responses were accepted, and response options included (1) “She/her”, (2) “He/him”, (3) “They/them”, (4) “Something else, including neopronouns such as ze/zir or fae/faer”, or (5) “Decline to answer.” Respondents were then coded into two groups: those who exclusively used she/her or he/him, and those who used they/them, neopronouns, or a combination of pronouns. 

### 2.3. Analyses

Chi-square tests of independence were used to examine differences in the proportions of youth who were able to change their identity documents based on the aforementioned sociodemographic and gender identity related covariates. Next, a multivariate adjusted logistic regression model was used to determine the adjusted odds of attempting suicide in the past 12 months among youth who had updated their documents compared to (1) youth who wanted to update their documents but were not able to and (2) youth who were able to update their documents but had not yet done so, after controlling for age, gender identity, sexual identity, race/ethnicity, income, U.S. region, immigrant status, pronoun usage, gender-affirming hormone status, and gender identity support from parents. SPSS 28 was used to conduct all analyses [17]. 

## 3. Results

### 3.1. Sample Description

A total of 11,914 youth from unique IP addresses indicated that they were transgender or nonbinary. The question about updating identification documents was near the end of the survey, and, as such, 2650 cases had missing data. Transgender and nonbinary youth who were not interested in changing their identification documents (*n* = 973) or did not know if they were interested in changing their identification documents (*n* = 1710) were excluded, resulting in a sample of 6581 transgender and nonbinary youth. Chi-square analysis of those with data on identification documents and without that data found that youth who had missing data about identification documents were more likely to be legal minors (59% vs. 55%, *X*^2^(1) = 8.09, *p* < 0.01), and White (61% vs. 55%, *X*^2^(5) = 41.3, *p* < 0.01), while those without missing data were more likely to be multiracial (21% vs. 17%, *X*^2^(1) = 41.35, *p* < 0.01).

In the analytic sample, the majority of youth identified as nonbinary (3627, 55.1%), followed by transgender boy/man (2346, 35.6%), and transgender girl/woman (608, 9.2%). The average age was 17.49 (SD = 3.18). Overall, 1314 (20%) identified as gay or lesbian, 128 (2%) as heterosexual, 1904 (29%) as bisexual, 1259 (19%) as queer, 1671 (25%) as pansexual, and 252 (4%) as questioning. The majority of the sample was non-Hispanic White (3535, 54%), followed by multiracial (1341, 20%), Latinx (790, 12%), Asian/Pacific Islander (305, 5%), Black (247, 4%), and American Indian/Alaskan Native (129, 2%).

### 3.2. Bivariate Analysis

A majority of respondents (3693, 56%) reported that they were not able to change their identification documents where they live but that they would like to if they were able to, 2428 (37%) reported that they were permitted to change their documents where they live and that they would like to but had not yet done so, and 460 (7%) reported that they had already changed their identification documents. Chi-square tests of independence revealed significant differences in official document changes by age, gender identity, sexual identity, race/ethnicity, Census region, pronouns, GAHT status, and support for gender identity (Table 1).

The proportion of youth who were able to change their identification documents among those who wanted to was highest for those ages 18–24 (109, 12%), those who identified as transgender boy/man (68, 11%) or transgender girl/woman (245, 11%), those whose sexual identity was straight (24, 19%), those who were White (298, 8%), and those who lived in the Northeast region of the U.S (109, 11%).

Among gender identity related covariates, a greater proportion of transgender and nonbinary youth who exclusively used binary pronouns (i.e., he/him or she/her) had updated their identification documents (283, 13%) compared to those who used nonbinary pronouns or combinations of pronouns (177, 4%). Transgender and nonbinary youth who had access to GAHT reported significantly higher rates of updating their official documents (364, 32%), compared to transgender and nonbinary youth who either did not want GAHT (45, 3%) or who wanted it but did not have access to it (42, 1%). There was also a significant relationship between parental support and having updated documents: transgender and nonbinary youth who reported that they had at least one parent or guardian who supported their gender identity reported higher rates of having updated their official documents (479, 27%), compared to youth who did not have a supportive parent or guardian (56, 3%) or youth who were not out about their gender (8, 0.5%).

In bivariate analyses, transgender and nonbinary youth who had changed their official documents reported lower rates of attempting suicide (49, 11%), compared to transgender and nonbinary youth who were not able to change their official documents (824, 24.5%) and transgender and nonbinary youth who were able but had not yet done so (438, 33.4%).

### 3.3. Multivariate Logistic Regression Analysis

After controlling for age, gender identity, sexual identity, race/ethnicity, socioeconomic status, U.S. Census region, immigration status, access to GAHT, parental support of gender identity, and pronoun usage, both wanting to update one’s documents but not being able to (aOR = 2.04, CI = 1.412–2.95; *p* < 0.001, Table 2) and being able to update one’s documents but not having done so (aOR = 1.74, CI = 1.22–2.50; *p* < 0.001) were associated with increased odds of attempting suicide in the last year, compared to youth who had changed their documents (Ref.) A Hosmer and Lemeshow test of goodness of fit was also insignificant (*X*^2^(7) = 6.73, *p* = 0.56), indicating that the model was a good fit.

## 4. Discussion

This study presents a novel analysis of the association between access to concordant identification documents and suicide risk among transgender and nonbinary youth. The findings indicate that both transgender and nonbinary youth who want to update their documents but cannot and youth who are able to update their documents but have not done so yet have significantly higher odds of attempting suicide in the past year, compared to youth who had updated their documents. These findings align with previous scholarship which has found that, among transgender adults, having at least one updated identification document was associated with lower prevalence of suicidal ideation and suicide attempts [10,11]. This scholarship suggests that decreasing barriers to updating one’s identification documents may decrease distress and stigma for transgender and nonbinary youth. Given that transgender and nonbinary youth are at a significantly elevated rate of suicide compared to their cisgender peers, these findings offer novel perspectives on suicide prevention among this high-risk population [5,18]. Both discrimination and harassment resulting from using non-concordant documents and the logistical challenges to updating one’s documents may be sources of minority stress for transgender and nonbinary youth [4]. Making identification documents easier to update and safer to use for transgender and nonbinary youth may decrease that minority stress and improve their suicide risk.

Transgender and nonbinary youth are not a monolith, and different members of the community reported different rates of being able to update their documents. Youth who used binary pronouns, youth who had access to GAHT, and youth who reported that they had at least one parent or guardian who was supportive of their gender identity all reported higher rates of having updated their identification documents. This may reflect the fact that youth who use binary pronouns have more opportunities to update their documents, with few jurisdictions offering nonbinary gender marker designations [6]. Indeed, because nonbinary gender markers can “out” document-holders as nonbinary or intersex, and given the growing number of terms used to self-identify gender, the safest and most inclusive policy would be to remove gender markers from identification documents. In 2019, the American Medical Association recommended removing sex designation from the public-facing portion of babies’ birth certificates, noting that the existing binary distinctions of male or female fail “to recognize the medical spectrum of gender identity” [19].

Similarly, access to GAHT and support from parents are closely related, since youth under the age of 18 require parental consent both to access GAHT and to change their documents [20]. Therefore, youth whose parents are supportive of their gender identity may have more access to GAHT and updating their identification documents. Additionally, in some jurisdictions medical intervention is required in order to change a gender marker, bolstering the association between access to GAHT and updating identification documents.

Transgender and nonbinary youth of color were also less likely to report having updated their identification documents compared to their White peers, which may be due to systematic barriers arising at the intersection of racism and transphobia, including distrust of the legal system, lack of access to bureaucratic resources, or lack of funds for legal and administrative fees.

### Limitations

This study uses a cross-sectional design and cannot be used to infer causation or directionality. There is a chance of reverse causation, with transgender and nonbinary youth who have better mental health being more equipped to navigate the process of updating documents, which can be onerous [21]. While the survey included validity checks (one looking for concordance between the same item placed at two points in the survey and one asking youth to select a specific response), to protect data quality, the data are still reliant on accurate self-reports from transgender and nonbinary youth. Finally, this study did not assess the barriers that youth encounter in trying to update their identification documents. Given the complexity of changing identification documents in various jurisdictions, youth may not have accurate information about whether or not they are able to change their documents. 

More research is needed to determine the exact barriers that prevent transgender and nonbinary youth from updating their identification documents: cost, lack of information, bureaucratic hurdles, etc. Additional research and support should be dedicated to youth who are immigrants, whose access to updated documents may be impacted by international law and their documentation status. Future research should consider the range of policies regarding updating identification documents, the myriad hurdles that transgender and nonbinary youth are forced to navigate during that process, and the variety of outcomes that transgender and nonbinary youth may experience due to incongruent documents, including educational attainment, employment, and housing.

## 5. Conclusions

These findings fill a gap in the literature about the association of access to concordant identification documents with the mental health of transgender and nonbinary youth and highlight the need to modernize procedures for updating all forms of government identification. Having documents that accurately reflect a person’s gender identity may decrease that person’s suicide risk and vulnerability to harassment and violence. Policy makers and document-issuing agencies should decrease barriers to updating one’s name and update, or remove, gender markers to promote better mental health among transgender and nonbinary youth.

## Figures and Tables

**Table 1 ijerph-19-05016-t001:** Sample characteristics stratified by official document change status.

	Not Able to, but Would Like to Change Them (%)	Yes, and Want to Change Them When Am Able (%)	Yes, and Has Changed Them (%)	*X*^2^ *(df) p*-Value
Age (*n* = 6581)				*X*^2^(2) = 449.07, *p* < 0.001
13–17	248 (66.1)	1163 (31.0)	109 (2.9)	
18–24	1209 (42.8)	1265 (44.8)	351 (12.4)	
Gender Identity (*n* = 6581)				*X*^2^(6) = 342.11, *p* < 0.001
Transgender boy/man	227 (37.3)	313 (51.5)	68 (11.2)	
Transgender girl/woman	1100 (46.9)	992 (42.3)	245 (10.8)	
Nonbinary assigned male at birth	231 (58.2)	145 (36.5)	21 (5.2)	
Nonbinary assigned female at birth	2135 (66.1)	978 (30.3)	117 (3.6)	
Sexual Identity (*n* = 6528)				*X*^2^(10) = 87.35, *p* < 0.001
Gay or lesbian	698 (53.1)	492 (37.4)	124 (9.4)	
Straight	45 (35.3)	59 (46.1)	24 (18.8)	
Bisexual	1074 (56.4)	713 (37.4)	117 (6.1)	
Queer	677 (53.8)	476 (37.4)	106 (8.4)	
Pansexual	1000 (59.8)	594 (35.5)	77 (4.6)	
I am not sure	164 (65.5)	77 (30.6)	10 (4.0)	
Race/Ethnicity (*n* = 6347)				*X*^2^(10) = 71.04, *p* < 0.001
African American/Black	169 (68.4)	73 (29.6)	5 (2.0)	
Asian Pacific Islander	190 (62.3)	99 (32.5)	16 (5.2)	
Latinx	497 (62.9)	248 (31.4)	45 (5.7)	
Native/Indigenous	82 (63.6)	42 (32.6)	5 (3.9)	
Multiracial	762 (56.8)	504 (37.6)	75 (5.6)	
White	1851 (52.4)	1386 (39.2)	298 (8.4)	
Income (*n* = 5901)				*X*^2^(2) = 1.67, *p* = 1.67
Just meeting basic needs	2288 (54.1)	1610 (38.1)	330 (7.8)	
More than meets basic needs	932 (55.7)	623 (37.2)	118 (7.1)	
U.S. Region (*n* = 6581)				*X*^2^(6) = 162.09, *p* < 0.001
Northeast	464 (47.6)	401 (41.2)	109 (11.2)	
South	1508 (64.4)	724 (30.9)	108 (4.6)	
Midwest	856 (58.3)	518 (35.4)	93 (6.3)	
West	867 (48.1)	785 (43.6)	150 (8.3)	
Immigrant Status (*n* = 6562)				*X*^2^(2) = 2.65, *p* = 0.27
Born in the U.S.	3499 (56.0)	2319 (37.1)	432 (6.9)	
Born outside of the U.S.	178 (57.1)	106 (34.0)	28 (9.0)	
Pronouns				*X*^2^(2) = 364.36, *p* < 0.001
Binary	873 (41.3)	959 (45.3)	283 (13.4)	
Outside of the binary	2816 (63.2)	1466 (32.9)	177 (4.0)	
GAHT Status (*n* = 6415)				*X*^2^(10) = 1657.41, *p* < 0.001
No, and don’t want them	1108 (69.4)	443 (27.8)	45 (2.8)	
No, but I would like them	2254 (61.3)	1383 (37.6)	42 (1.1)	
Yes	217 (19.0)	559 (49.0)	364 (31.9)	
Gender Identity Support (*n* = 6315)				*X*^2^(4) = 693.89, *p* < 0.001
No	1183 (64.3)	602 (32.7)	56 (3.0)	
Yes	1079 (39.5)	1265 (46.3)	479 (27.5)	
Not out about gender	56 (3.0)	388 (14.2)	8 (0.5)	
Outcome Variable				
Attempted suicide	824 (62.9)	438 (33.4)	49 (3.7)	*X*^2^(2) = 54.01, *p* < 0.001

**Table 2 ijerph-19-05016-t002:** Multivariate logistic regression results on the association between updating identification documents and likelihood of attempting suicide in the last 12 months.

		EXP(B)	95% C.I. for EXP(B)	
			Lower	Upper
Age	Age dichotomized into minors and above	2.28 ***	1.94	2.69
Gender Identity	Transgender girl/woman (Ref)			
	Transgender boy/man	0.98	0.75	1.28
	Nonbinary assigned male at birth	0.58 *	0.38	0.89
	Nonbinary assigned female at birth	0.81	0.59	1.11
Sexual Identity	Gay or lesbian (Ref)			
	Straight or heterosexual	0.87	0.49	1.52
	Bisexual	1.10	0.89	1.35
	Queer	0.93	0.73	1.18
	Pansexual	1.27 *	1.03	1.57
	Questioning	1.30	0.89	1.91
Race/Ethnicity	White (Ref)			
	American Indian/Alaskan Native	2.32 ***	1.48	3.64
	Asian American/Pacific Islander	1.05	0.73	1.52
	Black	1.39	0.97	2.01
	Latinx	1.16	0.92	1.46
	Multiracial	1.44 ***	1.21	1.72
Income	Just meets basic needs	1.86 ***	1.59	2.17
U.S. Region	Northeast (Ref)			
	South	1.09	0.87	1.36
	Midwest	0.98	0.77	1.25
	West	1.02	0.81	1.27
Immigrant Status	Born in the US or US territory	0.94	0.66	1.33
GAHT Status	Are you currently taking gender affirmation hormones?	1.16 *	1.01	1.34
Gender Identity Support	No supportive parent (Ref)			
	At least one supportive parent	0.71 ***	0.60	0.84
	Not out to parents about gender identity	0.60 ***	0.49	0.73
Pronouns	Use of nonbinary pronouns	1.34 **	1.08	1.66
Identification Document Status	Has changed legal documents (Ref)			
	Wants to change documents but is not able	2.04 ***	1.41	2.95
	Is able to change documents but has not	1.74 **	1.22	2.50
	Constant	0.06 ***		

* *p* < 0.05; ** *p* < 0.01; *** *p* < 0.001.

## Data Availability

In accordance with The Trevor Project’s data governance policies, raw data from this study are not available to outside researchers.

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
