# Peer review of "Association of Updating Identification Documents with Suicidal Ideation and Attempts among Transgender and Nonbinary Youth"

_ijerph, 2022, doi:10.3390/ijerph19095016_

Round 1
Reviewer 1 Report
See attached document for review notes.

Reviewer 2 Report
Dear Authors,
You have a topic very important.
I observ that is very important, for a part of this non-binary population, to have documents to conferm their desired gender. This fact is sufficient, for a part of them, for their process of gender adjustment with cross-sex therapy, for other also without therapy or surgery, demolition Vs reconstructive versus a consevative surgery.
I suggest this articol:
Ayden I Scheim, Amaya G Perez-Brumer, Greta R Bauer,
Gender-concordant identity documents and mental health among transgender adults in the USA: a cross-sectional study,
The Lancet Public Health, Volume 5, Issue 4, 2020, Pages e196-e203, ISSN 2468-2667, https://doi.org/10.1016/S2468-2667(20)30032-3.
Best regards

Reviewer 3 Report
It is a very interesting and necessary subject to study. But the manuscript has many shortcomings:
1. In line 33 the reference number is not in superscript, restify it. There are several more like this in the text, change them too.
2. In the sentence "A nonprobability sample of 34,759 lesbian, gay, bisexual, transgender, or queer 79 (LGBTQ) youth ages 13-24 who resided in the United States was collected online from 80 October 2020 to December 2020." Introduce some expression to better explain the parentheses, such as "A nonprobability sample of 34,759 people who declare themselves lesbian, gay, bisexual, transgender, or queer (LGBTQ) youth ages..."
3. The wording of the methodology and the summary lead to confusion. There is talk of orientation, but this does not influence the identification documents, since it is not included in them. Modify all that part.
4. Better explain the selection of the sample. How is the process to reach transgender and non-binary people.
5. Make a flowchart with the sample selection process in results.
6. In the identification documents, why are the other categories eliminated and dichotomized into two? Explain the reason for this union.
7. For suicide risk data, it should be better explained in the scale on which they are based. If it is validated, the population in which it is validated and its data.
8. Explain how many items were made for each of the variables.
9. The analysis section is poorly written. Write it better.
10. Ethical aspects are missing.
11. The results speak of significance but all the data is given in percentages. Give the p's and the results of the statistics.
12. In table 1 it is necessary to indicate what is shown inside the parentheses.
13. In table 2 or create another table to put the data of all the variables introduced in the model. It must be corrected that there are numbers without the zero in front of the "." In the table. Variance inflation factor data and calibration using the Hosmer-Lemeshow goodness-of-fit test are missing.
14. Reference 15 does not contribute anything.
15. The discussion must be broader and the references other than those used during the text.
16. There should be more references to support these findings.
Reviewer 4 Report
This is a well-written and structured article on the relationship between changing identification documents to match one’s gender identity and suicide ideation and suicide attempt. The findings are very useful for policy makers and could help form suicide preventive strategies. However I have some minor comments aimed at improving the manuscript.
Introduction
- In the last paragraph of this section the authors mention that this study aims to fill a literature gap on the subject. However, this is not previously addressed or explained.
- In this section it is mentioned that in this study many of the same variables as Scheim and colleagues’ 2020 study are going to be used. Yet, this is never explained as to why the authors selected Scheim’s study. I suggest this is explained in more detail.
- This section could benefit by including some parts of a more “traditional” introduction structure that includes the current status of the topic globally, the current status of the topic in the United States and the limitations of the current knowledge discussed.
- The main goal of the paper does not include where it is being done. Is it the whole US or only some states? Also the years that the study addresses are important because it could be pre, during or after Covid-19, which by itself had a big impact in mental health and suicide prevalence in the world.
Methods
- Why do the authors select the age group 13 to 24 years? It is not explained why? Is it because they are at more risk of suicide? I suggest this is justified in the introduction and explained in this section.
- The identification documents section includes the questions used in the survey. However when the authors merge two answers in the same category of not changed their documents (coded as 0) people who have not been able to change their documents and people that are able but have not done it (maybe because they are too young) this groups should refer to different matters. The first one could indicate that they are unable to change their documents, perhaps as they live in a more restrictive environment and context, which could put them in more risk of suicide than those in the second category. I suggest that a sensibility analysis is made using these responses separately in a multinomial logistic regression (with three categories: Not able, Able but has not done it, Able but has changed documents). This could rule out if there is a different suicide risk between all the categories of identification documents. As a matter of fact, this could be the main regression model of the paper. This is further strengthened by the results presented in table 1, where the identification documents categories are the three I just mentioned and the prevalence of the outcome variables between the three are significantly different.
Results
- Have the authors carried out a goodness-of-fit analysis of the logistic regression? It is not clear. Have they also checked for possible collinearity in the model mostly because as they state: “access to GAHT and support from parents are closely related, since youth under the age of 18 require parental consent both to access GAHT and to change their documents” or “youth whose parents are supportive of their gender identity may have more access to GAHT and updating their identification documents… bolstering the association between access to GAHT and updating identification documents”. I also suggest consider limiting the analysis to young people 18 or more, given that in some cases the access GAHT and to change their documents is limited by parental consent not because of the context, which could not be related to harassment and violence and thus could skew the results.
Round 2
Reviewer 1 Report
Thank you for making modifications based on my review. I don't have any further suggestions.
Reviewer 2 Report
Dear Autors,
I agree with the changes you made.
Best regards
Reviewer 3 Report
The quality of the manuscript has greatly improved with the changes made. But more changes are needed.
It is not indicated how the clusters were filled, or the size of the clusters.
They did remove the people who didn't want to change the documents because they no longer needed it, but they should have removed those who already had it because they no longer needed it either. inconsistent methodology.
In the pronoun variable, why do they eliminate those who did not want to answer?
In the Bivariate Analysis point they write "Overall, 3,693 (56%) of respondents...", they cannot put overall if it is half of the sample.
In the following paragraph "The proportion of youth who were..."; the chi square test shows that there are differences but not to which subgroups or between which the differences are. This paragraph makes no sense. This differentiation only makes sense when they are dichotomous variables.
In "Multivariate Logistic Regression Analysis", they put p=0.56 and give it as significant. The p must be p<0.5.
Reviewer 4 Report
The regression model should not be "Logistic regression" as in table 2 or in section 3.3 Multivariate Logistic Regression Analysis, or in the Methodology. It should be changed to Multinomial Regression Analysis.
